# Upper-Limb Isometric Force Feasible Set: Evaluation of Joint Torque-Based Models

**Nasser Rezzoug** [1,2,*] **, Vincent Hernandez** [3] **and Philippe Gorce** [1]

1   Faculty of Sports Sciences, Université de Toulon, CS 60584, 83041 Toulon, France; gorce@univ-tln.fr
2   Inria, Centre Bordeaux Sud-Ouest, Equipe Projet AUCTUS Inria/IMS (Univ. Bordeaux, CNRS UMR5218), 33405 Talence, France
3   Department of Mechanical Systems Engineering, Tokyo University of Agriculture and Technology, Tokyo 183-8538, Japan; vincent.hernandez1985@gmail.com
*   Correspondence: rezzoug@univ-tln.fr

**Abstract:** A force capacity evaluation for a given posture may provide better understanding of human motor abilities for applications in sport sciences, rehabilitation and ergonomics. From data on posture and maximum isometric joint torques, the upper-limb force feasible set of the hand was predicted by four models called force ellipsoid, scaled force ellipsoid, force polytope and scaled force polytope, which were compared with a measured force polytope. The volume, shape and force prediction errors were assessed. The scaled ellipsoid underestimated the maximal mean force, and the scaled polytope overestimated it. The scaled force ellipsoid underestimated the volume of the measured force distribution, whereas that of the scaled polytope was not significantly different from the measured distribution but exhibited larger variability. All the models characterized well the elongated shape of the measured force distribution. The angles between the main axes of the modelled ellipsoids and polytopes and that of the measured polytope were compared. The values ranged from 7.3° to 14.3°. Over the entire surface of the force ellipsoid, 39.7% of the points had prediction errors less than 50 N; 33.6% had errors between 50 and 100 N; and 26.8% had errors greater than 100 N. For the force polytope, the percentages were 56.2%, 28.3% and 15.4%, respectively.

**Keywords:** force feasible set; upper-limb; isometric torque; force polytope; force ellipsoid

## 1. Introduction

The knowledge of the maximum forces that an individual can exert on the environment allows one to assess his ability to perform a given task without exceeding physiological limits. In relation to the required force level, it is an important factor to consider when evaluating a workplace using an ergonomics assessment method such as RULA [1,2], REBA [3] or OCRA [4]. Maximum force can be used as an objective indicator for defining criteria for discomfort evaluation [5]; and in the framework of digital human modeling, it is needed for tuning muscle fatigue models' parameters [6]. Furthermore, it can contribute to a more efficient human–robot collaboration in manufacturing [7,8] or in rehabilitation contexts [9]. Indeed, if a robotic system is aware of the limits of the human operator or patient, it can evaluate more efficiently the assistance it needs to provide. Under a static condition, the distribution of the maximum external forces that the upper-limb can apply to the outside world, defined as the force feasible set (FFS), is known to be anisotropic and posture-dependent [10,11]. The dependence of the FFS on the posture is due to the muscle length, which determines the muscle's isometric force; muscular moment arms, which contribute to the joint torques; and the Jacobian matrix of the upper limb, which links these joint torques to the external force at the hand. Additionally, due to the anisotropy of the FFS, external force amplitude can be very different according to the direction of these forces. Therefore, the knowledge of a maximal value for a limited set of directions, as is usually found in the literature [12–15], may not be sufficient to globally represent the force

capabilities of an individual. It is believed that existing formalisms used in robotics could be interesting for evaluating the FFS of the hand in a given upper-limb posture because they have the ability to characterize the FFS in every Cartesian direction given the posture and assumptions of maximum joint torques. Two variants are available: the force ellipsoid (FE) [16] and the force polytope (FP) [17]. In the biomechanics field, the FE and FP can be adapted by considering measured maximum joint torques [11,18,19]. To apply these formalisms to human limbs, it must be noted that the maximum joint torques change as a function of joint angle and direction of rotation [20–22], and also depend on the other joint posture because of the presence of multi-articular muscles. The models based on measured joint torques are called the scaled force ellipsoid (SFE) and scaled force polytope (SFP) [11,19]. Considering the human upper-limb, few studies that were only conducted in the horizontal plane compared the SFE and SFP with measured forces [10,11,18]. For different elbow flexion values, a comparison between FFS predictions with FE, SFE, FP and SFP was done, but no ground truth measures of maximal forces exerted at the hand was proposed [23]. Additionally, some preliminary results on one subject with both joint torques and 3D force measurements were provided in [24]. In this framework, one purpose of the present study was to evaluate the four models of FFS (FE, SFE, FP and SFP) from recordings of human upper-limb posture and maximum joint torques with a dedicated dynamometer. The second purpose was to compare the modeled FFSs to a measured force polytope (MFP) constructed from the measurement of maximal forces in a set of directions obtained from a force sensor. Different parameters were considered in order to compare modeled FFSs and the MFP. They characterize the force amplitudes (maximum predicted force and volume which produces an overall evaluation) but also the shapes of the FFSs (more or less elongated) and their orientations (angle between the main axes of the FFS). In addition, the RMS error was calculated for all points on the surface of the modeled FFS and the MFP. It was hypothesized that SFE would underestimate the MFP, and the SFP would overestimate it thanks to their assumptions. It was also hypothesized that shape and orientation would be correctly predicted by the models.

## 2. Material and Methods

### 2.1. Upper-Limb Model

The considered model of the upper-limb has three segments (upper-arm, lower-arm and hand) and $n = 7$ degrees of freedom. The dimension of the end effector space is $m = 3$. The global reference frame and the frame assigned to each segment in the anatomical reference posture (the upper-limb along the body and palms of hands facing forward) is as follows: the X axis points towards the front of the body, the Y axis is vertical and the Z axis points laterally to the right. The longitudinal axis of each segment points in the direction of its Y axis. The glenohumeral joint (named hereafter the shoulder joint for simplicity) linking the upper-arm to the trunk is considered as a spherical joint with three degrees of freedom (dofs) (flexion–extension around the Z axis, abduction–adduction around X and internal–external rotation around Y). The elbow joint has two dofs corresponding to flexion–extension around Z and pronation–supination (Y). Finally, the wrist joint has two dofs corresponding to radial–ulnar deviation (X) and flexion–extension (Z). The upper-limb Jacobian matrix $J$ maps joint velocities $\dot{q} \in \mathbf{R}^n$ to end effector velocity $\dot{x} \in \mathbf{R}^m$, $\dot{x} = J\dot{q}$. $\mathbf{R}^m$ and $\mathbf{R}^n$ are the sets of real numbers of dimension $m$ and $n$, respectively. $J$ depends on joint angle vector $q \in \mathbf{R}^n$ and segments length. The elements $J_{ij}$ of $J$ are the partial derivatives of the end-effector coordinate $x_i$ relative to the joint angle $q_j$. In order to compute the Jacobian matrix needed for FFS computation, the seven-dofs kinematic chain model was defined through the Denavit–Hartenberg parameters with the sequences XZY for the shoulder, ZY for the elbow and XZ for the wrist. The elbow carrying angle (the angle between the upper-arm and lower-arm in the sagittal plane in the anatomical reference posture) was integrated into the upper-limb model but not considered as a dof (Table 1). Given the model and the joint angles obtained from inverse kinematics (detailed in Section 2.5), the kinematic chain Jacobian matrix was computed with the Matlab Robotics Toolbox [25].

**Table 1.** Upper-limb kinematic chain Denavit–Hartenberg parameters. S-FE/EX: shoulder flexion–extension; S-AB/AD: shoulder abduction–adduction; S-LR/MR: shoulder lateral–medial rotation; E-FE/EX: elbow flexion–extension; CA: elbow carrying angle; E-SU/PR: forearm pronation–supination; W-RD/UD: wrist radio–ulnar deviation; W-FE/EX: wrist flexion–extension; $L_u$: upperm arm length, $L_a$: lower arm length; $L_h$: distance from the wrist center to the handle.

| dof | $\alpha_i$ | $a_i$ | Offset | $d_i$ |
|---|---|---|---|---|
| S-FE/EX | $\frac{\pi}{2}$ | 0 | 0 | 0 |
| S-AB/AD | $\frac{\pi}{2}$ | 0 | $\frac{\pi}{2}$ | 0 |
| S-LR/MR | $\frac{\pi}{2}$ | 0 | $\frac{\pi}{2}$ | $-L_u$ |
| E-FL/EX | $-\frac{\pi}{2} + \text{CA}$ | 0 | 0 | 0 |
| E-SU/PR | $\frac{\pi}{2}$ | 0 | $\frac{\pi}{2}$ | $-L_a$ |
| W-RD/UD | $-\frac{\pi}{2}$ | 0 | $\frac{\pi}{2}$ | 0 |
| W-FL/EX | 0 | $-L_h$ | 0 | 0 |

*2.2. Force Ellipsoids and Polytopes Computation*

The aim of this section is to present the definitions of the force ellipsoids and polytopes. For each type, two variants are considered. The first one called simply FE or FP corresponds to unitary torque boundaries. The second variant, called SFE or SFP, considers experimentally measured maximum joint torques and their definitions. The capacities of FE and FP to predict maximum force directions and of SFE and SFP to predict direction and maximal force amplitudes were tested.

2.2.1. Force Ellipsoids Definition

The relation between the static force $f \in \mathbf{R}^m$ at the end-effector and joint torque $\tau \in \mathbf{R}^n$ is:

$$J^T f = \tau \tag{1}$$

The superscript $T$ represents the transpose of the corresponding matrix. The FE is the set of forces that can be exerted at the end-effector given that the joint torque type 2 norm is bounded by 1, i.e., if $\tau$ satisfies the inequality $\tau^T \tau \leq 1$. FE is defined as follows:

$$\mathcal{E}_f = \left\{ f \in \mathbf{R}^m \quad | \quad J^T f = \tau, \quad \tau^T \tau \leq 1 \right\} \tag{2}$$

Combined with (1), the definition of the FE becomes:

$$\mathcal{E}_f = \left\{ f \in \mathbf{R}^m \quad | \quad f^T J J^T f \leq 1 \right\} \tag{3}$$

In the case of a nonredundant 2-dof planar kinematic chain, the set of admissible joint torques is a circle (Figure 1A) that is mapped to an ellipse in force space through $J^{-T}$. In the case of a redundant chain such as the upper limb, the FE is the intersection between (3) and $Im(J^T)$ [17].

The SFE definition incorporates measured maximum isometric joint torques which are neither unitary nor symmetrical. In this case (3) becomes [11]:

$$\mathcal{SE}_f = \left\{ f \in \mathbf{R}^m \quad | \quad \left( f - \overline{f} \right)^T \hat{J} \hat{J}^T \left( f - \overline{f} \right) \leq 1 \right\} \tag{4}$$

with

$$\hat{J} = J T_\tau \tag{5}$$

$$T_\tau = diag \left( \frac{1}{\tau_1^{max} - \overline{\tau_1}}, \cdots, \frac{1}{\tau_n^{max} - \overline{\tau_n}} \right) \tag{6}$$

$$\overline{\tau_i} = \frac{\tau_i^{max} + \tau_i^{min}}{2} \quad i = 1, \ldots, n \tag{7}$$

$$\overline{f_i} = \frac{f_i^{max} + f_i^{min}}{2} \quad i = 1, \ldots, m \tag{8}$$

$\tau_i^{max}$ and $\tau_i^{min}$ represent the maximum torque in the positive and negative joint rotation directions, respectively.

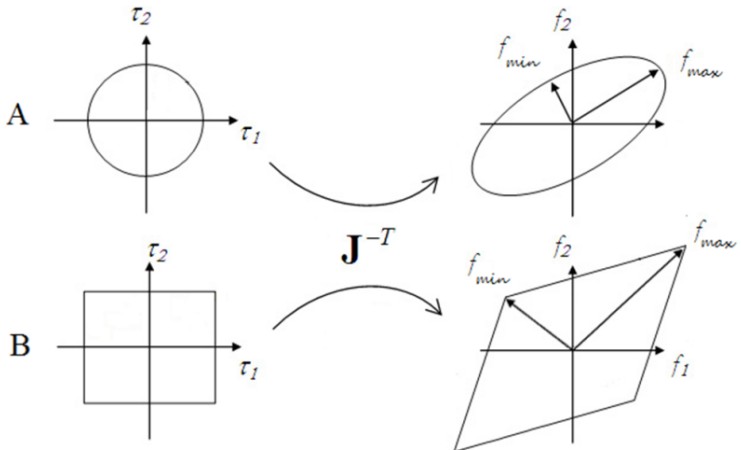

**Figure 1.** Definition of FE and the FP for a 2-dof nonredundant kinematic chain. (**A**) force ellipse for which the joint torques are bounded by a type 2 norm. The set of joint maximum torques is a circle that is mapped to an ellipse in the force space through $J^{-T}$. (**B**) Force polytope for which the joint torques are bounded by an infinite norm; the set of maximum joint torques is a rectangle that is mapped to a polygon in force space.

The FE or SFE main axes magnitude and orientation are obtained by applying a singular value decomposition (SVD) on $J$ and $\hat{J}$, respectively.

$$J = USV^T \tag{9}$$

The columns of the orthogonal $[m \times m]$ matrix $U$ contain the directions of the ellipsoid's principal axes in the force space. The columns of the orthogonal $[n \times n]$ matrix $V$ form an orthonormal basis of the torque space. The inverse of the corresponding singular values ($\sigma_1 > \sigma_2 > \sigma_3$) stored in $S$ (dim $S = m \times n$) provide the lengths of the ellipsoid axes. The volume of the ellipsoid is obtained by the following equation:

$$Volume = \frac{4\pi}{3\sigma_1\sigma_2\sigma_3} \tag{10}$$

The isotropy describes the global shape of the ellipsoid [26]. A value of 0 (isotropic condition) means that the maximal force is the same in all directions and that the FE or SFE is spherical; a value near 1 denotes a very elongated shape and the existence of a direction for which the maximum force is large. The isotropy is computed from $1/\sigma_3$ and $1/\sigma_1$, the maximal and minimal singular values, respectively, of $J^T$ or $\hat{J}^T$.

$$w = \sqrt{1 - \left(\frac{\sigma_3}{\sigma_1}\right)^2} \tag{11}$$

2.2.2. Force Polytopes Definition

The force polytopes definition is based on the assumption that the joint torques are bounded by an infinite norm. The FP and SFP definitions are given below:

$$\mathcal{P}_f = \{f \in \mathbf{R}^m | J^T f = \tau, \quad \tau_i \in [-1, 1] \quad i = 1, \ldots, n\} \tag{12}$$

$$\mathcal{SP}_f = \{f \in \mathbf{R}^m | J^T f = \tau, \quad \tau \in [\tau_{min}, \tau_{max}]\} \tag{13}$$

In the nonredundant 2D case the set of admissible torques is a rectangle that is mapped to a polygon in force space through $J^{-T}$ the inverse of the Jacobian matrix transpose (Figure 1B). $\tau_{min} \in \mathbf{R}^n$ (negative values) and $\tau_{max} \in \mathbf{R}^n$ correspond to the maximum isometric torques measured at each dof for each movement direction. The vertices of the FP and SFP sets can be obtained by resolving a set of linear equations with slack variables [17] or by using a geometrical interpretation [27,28]. For each polytope (FP and SFP), the magnitude and direction of the main axes are obtained from the SVD of their vertices and the volume is obtained numerically from their convex hull. Finally, the isotropy is determined from the singular values stored in the $S$ matrix (9).

### 2.2.3. Measured Force Polytope

To assess the validity of the modelled force ellipsoids and polytopes, an experiment was set up for measuring both maximum joint torques and maximum forces that constitute the input and output of the FFS models respectively. A measured force polytope (MFP) was defined by considering the convex hull of the maximum isometric forces recorded with a 3D force sensor. For comparison purposes, its main axis, volume and isotropy were assessed in the same way than those of FP and SFP. The experimental procedure is now described.

### 2.3. Participants

Seven right-handed male sport science students (20.0 (2.7) years old, 180.4 (7.7) cm and 80.4 (10.4) kg) without any upper limb pathology were recruited for the experiment. All subjects were fully informed about the procedures and aims of the experiment. Informed consent was obtained from all subjects involved in the study. The study was conducted according to the guidelines of the Declaration of Helsinki, and approved by the local Ethics Committee of the university of Toulon.

### 2.4. Materials

An OQUS 400 optoelectronic system (Qualisys AB, Gothenburg, Sweden) composed of 6 cameras was used to track at 200 Hz the Cartesian position of nineteen reflective markers placed on the upper-limb and the trunk in accordance with the International Society of Biomechanics norm [29]. Isometric joint torques were measured with a Biodex 3 system (Biodex Medical Systems, Shirley, NY, USA) with a sampling rate of 100 Hz. Finally, a triaxial force platform AMTI–1000 series (Advanced Mechanical Technology Inc., Watertown, MA, USA) equipped with a custom made handle was used to assess the force production at the hand at a sampling rate of 100 Hz. Force and Biodex data were synchronised by using the Qualisys USB Analog Acquisition interface 64 channels (Qualisys AB, Gothenburg, Sweden).

### 2.5. Experimental Protocol

During the force measurements, the posture of the participant was standardized. The shoulder was slightly flexed an abducted with an elbow flexion of around 70°, a pronated forearm around 80°. The wrist was slightly extended. This posture is very common and corresponds to many situations in the workplace such as drilling or cart pushing for example. Additionally, this choice was dictated by the experimental set up with both joint torque and force measurements which was rather constraining and did not allow us to test many different postures. The subject was sitted in the Biodex dynamometer chair with the trunk attached by safety belts (Figure 2). The experimental procedure was separated in two parts (one for the force measurements and another one for the joint torques measurements) with at least 48 h between them to avoid the effect of muscular fatigue. The evaluation of the MFP was done for one upper-limb posture.

The first day of the experiment, the maximal isometric force was measured at the hand in 26 directions for the considered upper-limb posture. These directions are described by their azimuth and elevation. Eight azimuths were considered from 0° (horizontal forearm

longitudinal axis (Figure 2A) to 315° with steps of 45° and 3 elevations : 0° (horizontal plane ), 45° (upward) and −45° (downward) completed with one trial at 90° of elevation (upward) and one at −90° (downward). The order of passage was defined randomly to avoid bias due to the fatigue. For each measurement, the participant was asked, with verbal encouragement, to produce the most important force in the proposed direction for 3 s. A rest period of 3 min was allowed between each trial. The second day, the maximum isometric joint torque at each of the 7 degrees of freedom (dofs) of the upper-limb in the standardized posture of the first day were measured: 3 dofs for the shoulder (flexion/extension, adduction/abduction, external/internal rotation), 2 dofs for the elbow (flexion/extension and supination/pronation) and 2 dofs for the wrist (radial/ulnar deviation and flexion/extension). Two trials of 4 s were performed for each movement direction at each dof. The greatest value among the two trials was considered as the maximum isometric torque. A rest period of 3 min was allowed between each trial. The upper-limb segments were placed with respect to the Biodex machine by using manufacturer recommendation. Moreover, the dynamometer axis was aligned with the considered joint center with fine adjustments (dynamometer height and the seat advancement). The posture of the participant during maximum torque measurements was adjusted to be as close as possible to that of the maximum force measurements. Additionally, the joint rotation centres relative to the external markers was determined by using the SCoRe methods [30] and used to compute the segments length. To do so, the subjects were instructed to perform full range movement of each dof for 10 s [31,32] at a freely chosen velocity [33]. Then, the misalignment with the dynamometer axis and the adjusted resultant moment around the considered joint axis were determined [34–37]. Moreover, gravitational effect was corrected before each measurement by the procedure defined by the dynamometer manufacturer. The posture of the upper-limb was obtained from nineteen light reflective markers placed on the trunk and on the right upper-limb fixed to the skin with a hypoallergenic adhesive tape. Thirteen markers were positioned on anatomical landmarks identified by palpation. They were placed as follows on: the xiphoid process, suprasternal notch, one on each most dorsal point on the acromioclavicular joint, lateral and medial epicondyles of the humerus, most caudal-lateral point on the radial styloid and most caudal-medial point on the ulnar styloid processes, middle of the third metacarpi, distal extremity of the second and fifth metacarpi, seventh cervical vertebra and eighth thoracic vertebra. Three additional technical markers were placed on the upper-arm and the forearm. In order to compute the Jacobian matrix needed for FFS computation, the length of the segments of the upper-limb model was obtained by computing the distance between the joint centers obtained from anatomical markers (middle of the radial and ulnar styloids for the wrist and middle of the medial and radial epicondyles for the elbow) and by the ScoRe method for the glenohumeral joint center [30,31]. The distance between the wrist and the handle ($L_h$) was approximated by that between the wrist center and the middle of the third metacarpi. The elbow carrying angle, assessed from a static posture in the neutral position, was integrated in the upper-limb model but not considered as a dof. The position of the technical markers in the frame of the segment on which they were attached was computed from a static posture similar to that of the force measurements. Then using the upper-limb model and the technical markers, the joint angles were computed by the inverse kinematics algorithm proposed in [38,39]. Given the model and the joint angles, the kinematic chain Jacobian matrix was computed with the Matlab Robotics Toolbox [25]. Then, for each subject, the maximum measured joint torques $\tau_{min} \in \mathbf{R}^n$ and $\tau_{max} \in \mathbf{R}^n$ for each dof and each movement direction were used to compute the SFE and SFP according to the methods described in Sections 2.2.1 and 2.2.2 [17], respectively.

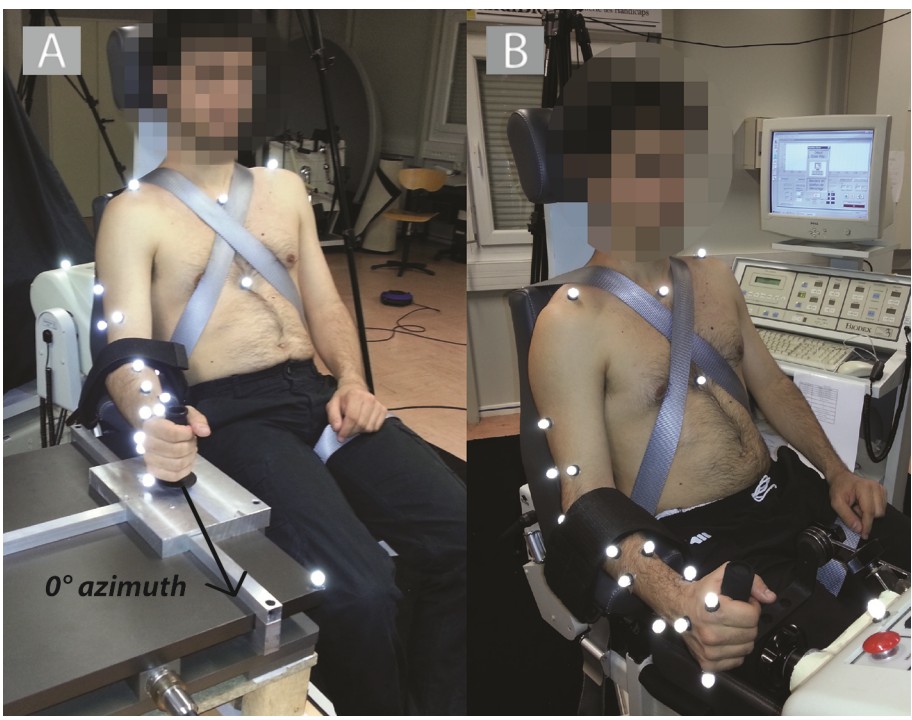

**Figure 2.** Experimental set up: (**A**) maximum force measurement. The subject grasps a handle rigidly attached to a force plate. The belt on the forearm is not tightened. (**B**) An example of maximum isometric torque measurement on the Biodex ergometer (wrist pronation–supination).

Data Analysis

After checking the normality of the data with the Shapiro-Wilk test, a one way Analysis of Variance (ANOVA) with repeated measures was performed. The dependent variables were: the maximum forces, the volume, the isotropy and the angle between the main axes of MFP relative to FE, SFE, FP and SFP. Then, Dunnett post hoc tests were used to compare the parameters of FE, FP, SFE and SFP with those of MFP or Tukey post-hoc tests for comparing the angles between the FFSs and MFP main axes. The significance level was set at 0.05 and the Statistica software (Statsoft, Tulsa, OK, USA) was used. In addition, to compare more globally the RMS error (RMSE) between the SFE and the SFP with the MFP, a color gradient sphere was considered. It was defined by a set of direction $\mathbf{v}(r, \theta, \varphi)$, with r = 1, $\theta \in [0, 360°]$ and $\varphi \in [0, 180°]$, whose spherical coordinates varied with an increment of 1° and calculated as follows:

$$\mathbf{v} = \begin{pmatrix} \cos\theta & -\sin\theta & 0 \\ \sin\theta & \cos\theta & 0 \\ 0 & 0 & 1 \end{pmatrix} \begin{pmatrix} \cos\varphi & 0 & \sin\varphi \\ 0 & 1 & 0 \\ -\sin\varphi & 0 & \cos\varphi \end{pmatrix} \begin{pmatrix} 1 \\ 0 \\ 0 \end{pmatrix} \tag{14}$$

According to each of these vectors, the RMS error between the intersection of the MFP with the SFE and SFP was determined. Thus, the graphical representation consisted of a sphere whose surface was colored according to the RMS error related to the **v** direction. For a sufficiently tight sampling of points, one could thus scan its entire surface and associate a color code to each point. In addition, the percentages of points for which the RMSE was less than 50 N; between 50 and 100 N; or greater than 100 N were determined. Finally, mean, minimum and maximum RMSE values were also provided. Finally, the upper-limb joint angles, were compared between the torque and force measurements with a student *t* test.

## 3. Results

In this section, the results of the comparison between the parameters of MFP and those of FE, FP, SFE and SFP are presented.

The measured joint torques are summarized in Table 2. The measured joint angles during the joint torques and force measurements were relatively close despite two significant differences (Table 3). The mean(SD) of joint angles displayed on the first line of Table 3 correspond to that of the torques measurement on the dynamometer for the considered degree of freedom. The second line provides the mean(SD) of the joint angles of the complete upper-limb dofs during the force measurements.

The ANOVA on isotropy showed significant main effect ($F(4, 24) = 8.40$, $p < 0.05$) and the post hoc test revealed lower value of isotropy for the MFP compared with the FE, SFE, FP and SFP (FE: $0.969 \pm 0.006$, SFE: $0.970 \pm 0.010$, FP: $0.971 \pm 0.007$, SFP: $0.966 \pm 0.015$ vs. MFP: $0.926 \pm 0.034$, $p < 0.05$ for each comparison) (Figure 3).

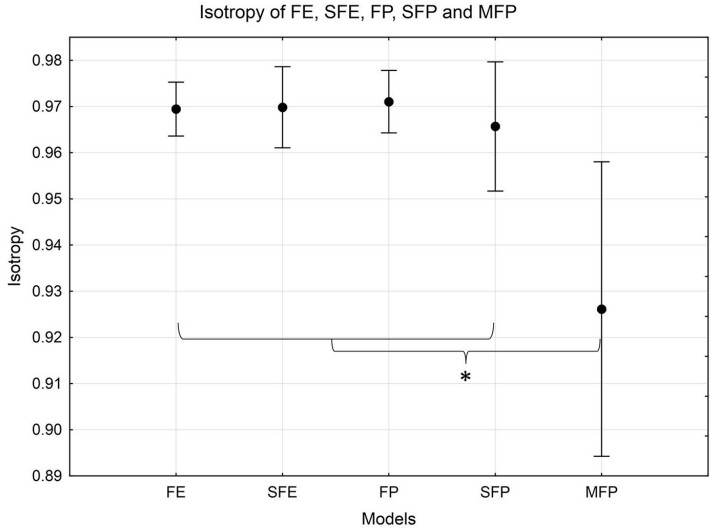

**Figure 3.** Isotropy of FE, SFE, FP, SFP and MFP (MFP is significantly different to all models, * $p < 0.05$).

The FE and FP volumes and maximal forces were not included in the statistical analysis because of the unitary joint torques assumption. The ANOVA on volume showed significant main effect ($F(2, 12) = 10.081$, $p < 0.05$) and post hoc test indicated that the SFE volume was lower than that of MFP (SFE: $1.3 \times 10^7 \pm 1.0 \times 10^7$ N$^3$ vs. MFP: $4.6 \times 10^7 \pm 2.9 \times 10^7$ N$^3$, $p < 0.05$) (Figure 4).

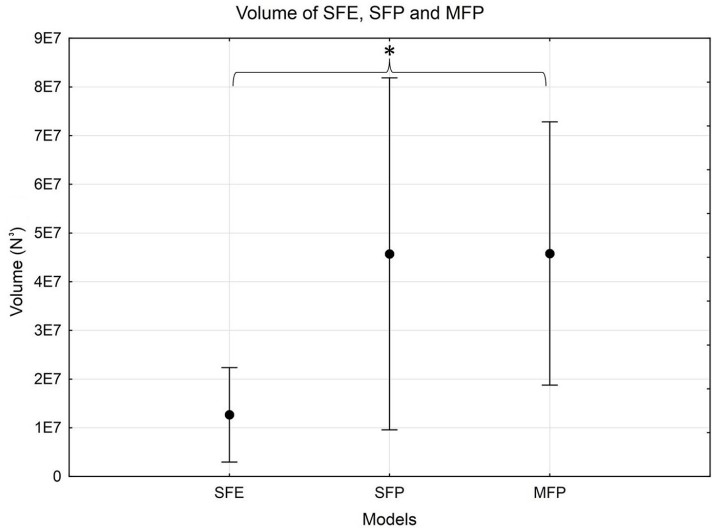

**Figure 4.** Volume (N$^3$) for the SFE, SFP and MFF (* $p < 0.05$).

**Table 2.** Mean (SD) measured joint torques (in N·m) at the shoulder, elbow and wrist joints. The sign convention is indicated in parentheses for each dof movement direction. Abbreviations: flexion (FL), extension (EX), adduction (AD), abduction (AB), lateral rotation (LR), medial rotation (MR), supination (SU), pronation (PR), radial deviation (RD), ulnar deviation (UL), (+) : counterclockwise rotation, (−) clockwise rotation.

| Shoulder | FL (+) 71.6 (17.0) | EX (−) 87.5( (25.7) | AD (+) 81.0 (25.0) | AB (−) 68.0 (19.9) | MR (+) 53.9 (12.0) | LR (−) 41.7 (5.8) |
|---|---|---|---|---|---|---|
| Elbow | FL (+) 68.1 (18.9) | EX (−) 61.0 (11.5) | PR (+) 9.3 (2.3) | SU (−) 11.8 (5.3) | | |
| Wrist | FL (+) 14.8 (8.3) | EX (−) 7.9 (3.9) | UD (+) 17.1 (8.0) | RD (−) 14.4 (7.1) | | |

**Table 3.** Mean (SD) joint angles (in degrees) obtained during joint torque measurements on the dynamometer and force measurements with the force sensor (* $p < 0.05$).

| | S-FE/EX | S-AB/AD | S-LR/MR | E-FL/EX | E-SU/PR | W-RD/UD | W-FL/EX |
|---|---|---|---|---|---|---|---|
| Dynamometer | 6.2 (4.7) * | −22.3 (5.4) | 5.7 (3.9) | 71.3 (6.7) | 75.1 (7.9) | 4.5 (5.2) * | −12.4 (3.5) |
| Force sensor | 12.9 (7.0) * | −22.2 (3.5) | −1.5 (7.9) | 66.5 (6.6) | 80.1 (6.4) | −3.0 (4.4) * | −18.5 (8.4) |

The ANOVA on maximal forces showed a significant main effect ($F_{(2, 12)} = 27.298$, $p < 0.05$). The Dunnett post hoc test indicated that the MFP maximal force (Figure 5) was lower than that of SFP and higher than that of SFE (SFE: 329.2 ± 79.5 N vs. MFP: 518.1 ± 134.4 N, $p < 0.05$; SFP: 619.0 ± 160.3 N vs. MFP, $p < 0.05$) (Figure 5).

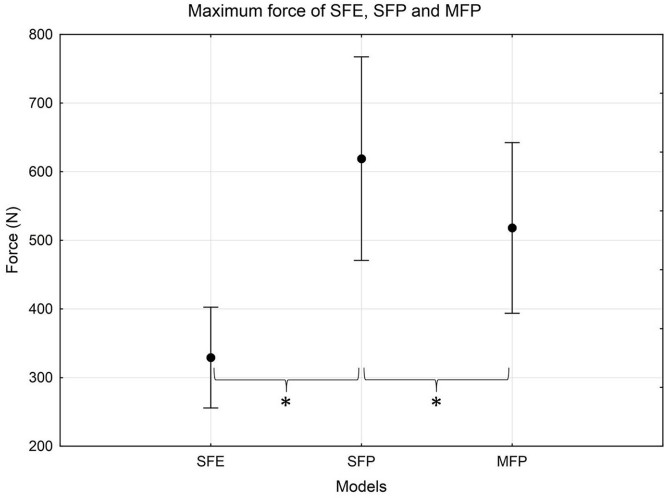

**Figure 5.** Maximum forces (N) for the SFE, SFP and MFP (* $p < 0.05$).

Results concerning the angles between the main axis showed a significant main effect ($F_{(3, 18)} = 14.192$, $p < 0.05$). Tukey's post-hoc indicated that both αSFE/MFP and αSFP/MFP are significantly smaller than αFE/MFP and αFP/MFP ($p < 0.05$) (Table 4).

**Table 4.** Mean angle (°) mean (SD) formed by the main axis of MFP with the FE, SFE, FP and SFP. (* $p < 0.05$, αSFE/MFP < αFE/MFP and αSFE/MFP < αFP/MFP; £ $p < 0.05$, αSFP/MFP < αFP/MFP and αSFP/MFP < αFE/MFP).

| αFE/MFP | αSFE/MFP | αFP/MFP | αSFP/MFP |
|---|---|---|---|
| 12.8 (3.1) | 7.4 (3.3) * | 14.6 (3.0) | 9.3 (5.3) [£] |

As an example, the sphere of RMS errors between MFP, SFE and SFP is given for a subject in Figure 6 and RMS errors data are provided in Table 5.

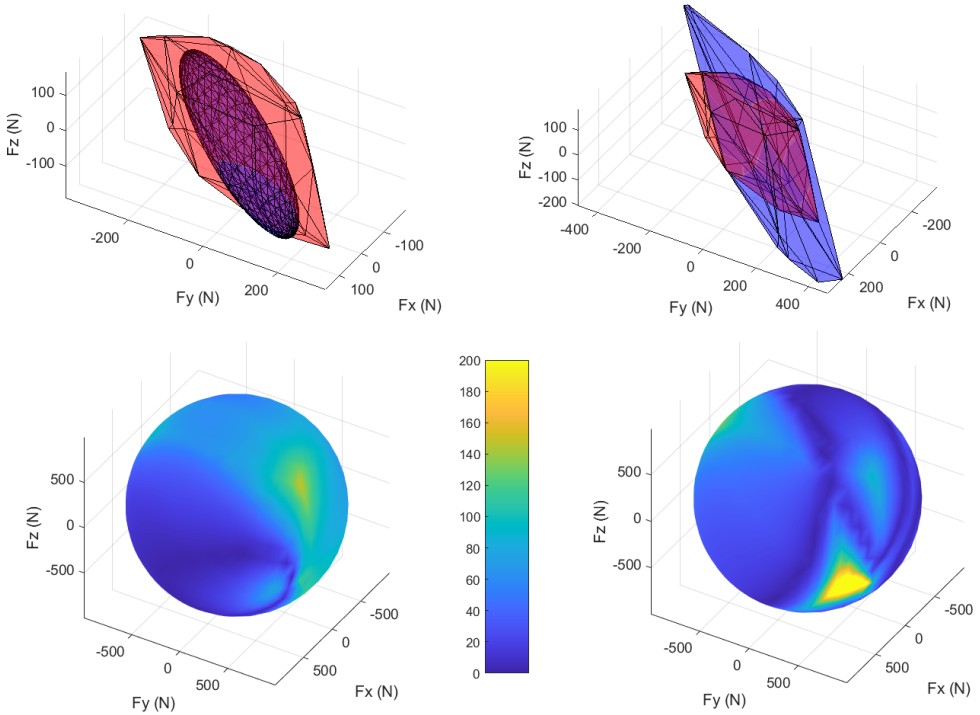

**Figure 6.** Comparison between the prediction models (SFE and SFP) and the MFP. (**upper left**) SFE and MFP are superimposed; (**upper right**) SFP with MFP; (**lower left**) RMS error sphere between SFE and MFP; (**lower right**) between SFP and MFP, the color code applies to both figures and goes from dark blue to light yellow in ascending order of RMS error; the values are given in N.

**Table 5.** Mean, minimum, maximum RMSE (standard deviation) and the percentages of the area of the spherical RMSE whose values are less than 50 N, between 50 N and 100 N or greater than 100 N—between the SFE and the MFP on the first line and between the SFP and the MFP on the second line.

|  | RMSE Mean | RMSE Min | RMSE Max | % X < 50 N | % 50 < X < 100 N | % X > 100 N |
|---|---|---|---|---|---|---|
| SFE vs. MFP | 72.8 (16.0) | 1.3 (3.2) | 244.0 (79.1) | 39.7 (12.0) | 33.6 (12.8) | 26.8 (13.0) |
| SFP vs. MFP | 56.4 (16.7) | 0.13 (0.13) | 305.6 (85.3) | 56.2 (14.0) | 28.3 (7.2) | 15.4 (8.7) |

## 4. Discussion

The objective of this study was to evaluate four prediction models of force feasible sets (FE, SFE, FP and SFP) against force measurements (MFP). Comparisons were made on various representative parameters such as global orientation, volume, isotropy and maximum force. In addition, detailed prediction errors were evaluated. These results are original and constitute the first comparison of 3D modeling of FFS with both hand force and joint torque measurements on the upper limb. As such, these results significantly complement and enrich previous studies that involved only a very small number of subjects [10,11,18] or only considered maximal force [40,41]. We will now comment on the differences observed in light of the data in the literature, while taking a critical look at the hypotheses retained for the models. In addition, our results allow us to discuss the relative importance of the postural component, i.e., that related to the Jacobian matrix, compared to that related to the joint torques.

Data for measured isometric maximum joint torques were consistent with literature data for similar postures and in the same population [20,22,42,43]. For example, shoulder joint torque values for extension and abduction for a healthy population aged 20–29 years are

91.9 (19.7) [42] N·m and 60.2 (14.0) N·m [22] compared to 87.5 (25.7) N·m and 68.7 (20.0) N·m in this study, respectively.

The study of the literature enabled us to highlight the anisotropy of the FFSs, i.e., the difference in the amplitude of the forces according to the direction of application [10,11]. All the proposed FFS models had this property. In fact, we can see that the isotropy values were higher than 0.93, characteristic of an elongated shape. Consequently, a preferential direction of the force was obtained, generally oriented in the antero-posterior and down-ward direction (Figure 7). The isotropy of the MFP was statistically lower than that of the four models. The more elongated shape of the SFE compared to the MFP was due to the fact that there was an underestimation of the forces along the smaller axes of the ellipsoid compared to the preferential axis. For the SFP, the opposite trend was observed with a correct estimate along the smaller axes and an overestimation along the major axis (Figure 8).

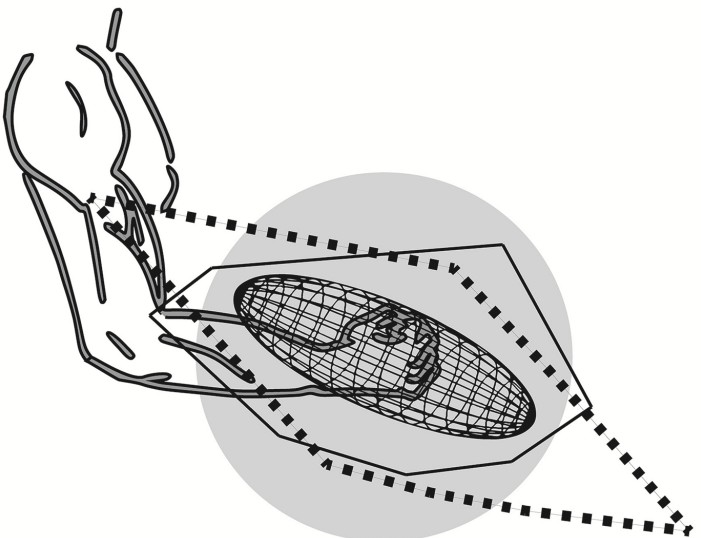

**Figure 7.** Representations of the SFE, SFP and MFP in the sagittal plane relative to the upper-limb posture for one subject, MFP (black squares), SFE (wire-frame) and MFP (solid line). The gray circle has a radius of 300 N.

The volume of the SFE significantly underestimated the overall measured force production capacities (MFP) whereas no statistical difference was found between the MFP and the SFP. For the latter, a significant inter-individual variability was observed and no clear trend appeared with respect to the MFP with, depending on the subjects, lower or greater values. It thus appeared that the assumptions chosen for the type of joint torques norm had a strong influence on the volume of the FFS models.

Moreover, compared to the MFP, there was a significant underestimation of the maximum force by more than 36.5% by the SFE and an overestimation of 19.5% by the SFP. Once again, the assumptions used to construct the models significantly affected the observed results. Indeed, the definition of the ellipsoids was based on the use of an Euclidian-type norm in order to define the joint torques limits. Thus, when a joint torque was maximum at the level of one dof, the others were all equal to zero, which was not very representative of an individual's motor control. In the musculoskeletal system, when a maximum joint torque is produced at the shoulder, it is possible to generate a joint torque at the elbow. Conversely, for polytopes, the infinite type norm allowed all torques to be at their maximum values simultaneously, which also seemed physiologically unrealistic. Thus, as expected, there was an underestimation of the FFS with SFE and an overestimation with SFP. The fact that a maximum torque on one dof leads to a null value on the others limits the force production evaluation. Similarly, the possibility that all torques are maximum does not

take into account the coupling between the different degrees of freedom. The latter is due to the presence of bi and pluri-articular muscles of the musculoskeletal system.

The parameter that was best evaluated by the models was the direction of the main axis of the MFP. In the case of ellipsoids, the maximum forces were exerted according to the latter. For polytopes, the SVD allowed one to find the principal axis taking into account globally all its vertices. Thus, this axis may not correspond exactly to the maximum forces. In the present case, the angle between the principal axis of the FE, the SFE, the FP and the SFP with that of the MFP was between $7.4 \pm 3.3°$ and $14.8 \pm 3.0°$ (absolute value: 2.9 to 19.6°) (Table 4). In terms of overall orientation, the biomechanical models (SFE and SFP) provided the best results compared to the MFP.

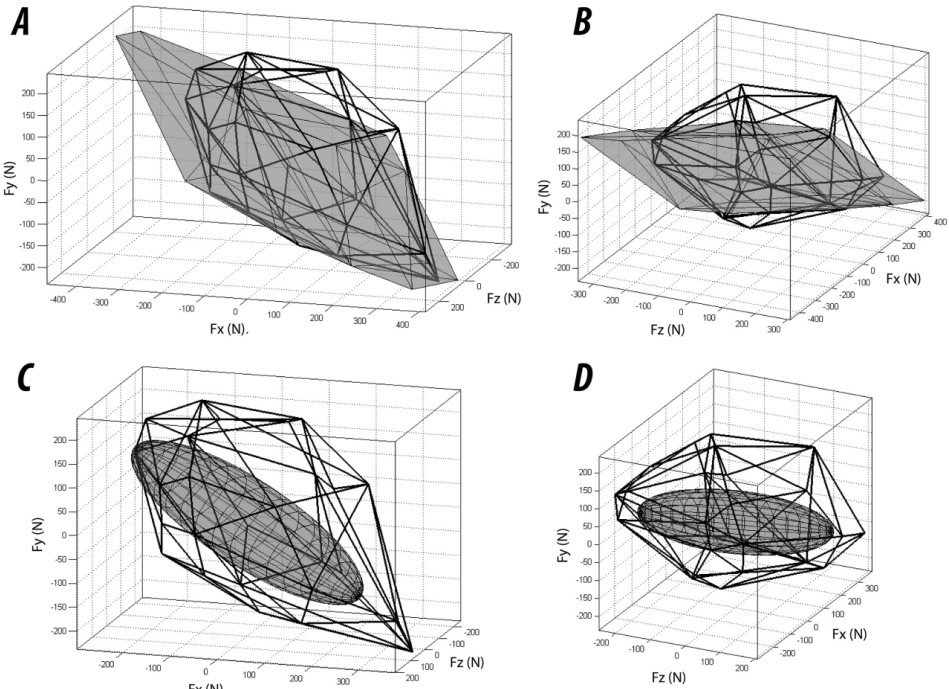

**Figure 8.** (**A**,**B**) Representation of the MFP (thick line) and the SFP (dark gray) for a subject from two different points of view. (**C**,**D**) Representation of the MFP (dark line) and the SFE (dark gray) for the same subject from two different points of view. Fx, Fy and Fz represent the forces for the anteroposterior-posterior axis, the vertical axis and the medio-lateral axis, respectively.

The characteristics of the proposed FFS models depend on three parameters: the posture of the kinematic chain represented by the Jacobian matrix, the joint torques and the types of norms used to express the limits of the latter. For the considered posture, the results suggest that the isotropy is little affected by the type of model, because the variations observed between them were small (between $0.966 \pm 0.015$ for the SFP and $0.971 \pm 0.007$ for the FP: $0.971 \pm 0.007$). Thus, one could deduce that for this characteristic of the FFSs, the postural component is preponderant. This observation is supported by [23], which showed that depending on the elbow flexion angle, isotropies vary significantly between nonscaled and scaled models. It was found that the incorporation of measured values of maximum joint torques (SFE and SFP) improved the prediction of the orientation of the FFS models with respect to the MFP. The appeal of such information is twofold [19,44,45]. The FFS provides information on the set of forces that the upper limb can exert on the environment. In particular, some directions allow for greater forces than others because of the anisotropy of the FFS. If for a given task corresponding to a given direction of force (e.g., drilling), the operator's posture causes the largest axis of the FFS to be aligned with the direction of the applied force; this might suggest that the posture is chosen so as to maximize the possible force in the direction of interest. In other words, it would mean that the operator has chosen his posture in such a way that he can exert the greatest possible force given the

capacities of his musculature. Moreover, the greater the force to be applied, the more the possible choices in terms of the combination of muscular forces should be reduced and the main axis of the FFS should be aligned with the force direction related to the task [19,45,46]. Based on this information, it may be possible to modify the operator's posture in order to maximize the transmission of force and thus limit the joint torques and reaction forces at the origin of potential musculoskeletal disorders [44]. The exertion of high intensity forces, handling heavy load over long period of time and working in unfavorable postures are main factors contributing to the incidence of musculoskeletal disorders. High joint torques may be associated with high muscular forces and increased joint reaction forces responsible for muscle and articular tissue damage leading to musculoskeletal disorders. While the results concerning FFS orientations seemed satisfactory, those related to the maximum amplitude of forces must be improved because of the significant variability of the SFP and the differences between the SFE and the MFP. The implementations of the FE and the FP are the simplest because they only require knowledge of the Jacobian matrix and therefore of the posture of the upper limb model. However, these two formalisms were the ones for which the orientation errors of the principal axis with respect to the MFP main axis were the most important. Moreover, they do not give any information on the amplitudes of maximum force. Their use could therefore be envisaged if the degree of precision required on orientation is less crucial (of the order of 15°). The SFE and the SFP provided more precise information and should be preferred. If we consider the results relating to volume and RMSE errors (Table 5), the SFP clearly appeared to be the best performing model, with, in particular, a percentage of points with an RMSE < 50 N significantly higher than that of the SFP (56.2 (12) vs. 39.7 (12.0)). However, the prediction error on maximum force was greater for the SFP compared to that of the SFE (305.6 (85.3) vs. 244.0 (79.1) N). Therefore, overall, the SFP was the model that seemed the most convincing.

This work has several limitations. Indeed, only one posture has been tested and this analysis should be extended to other cases. One of the limitations concerns the value of joint angles during the measurement of joint forces and torques. Indeed, the configuration of the dynamometer did not always allow one to have joint angles of the unmeasured dofs very close to those adopted during the force measurement. For example, during shoulder torque measurements it was not possible to flex the elbow sufficiently. Since the maximum torques may depend on the position of adjacent joints, this could have had an effect on the torque value. However, special care was given to having postures as close as possible for the two types of measures. In addition, it is necessary to validate the predictions on a wider range of subjects, especially those with pathologies following trauma (spinal cord injury) or central nervous system damage (stroke). It will then be possible to test whether these models are relevant for detecting impairments [46,47]. The evaluation of MFP could be improved by providing 3D visual feedback of applied forces to allow better control of the direction of force application. Model scaling is an important issue in the field of biomechanics, and the evaluation of joint torques requires a long and tedious experimental protocol. This temporal constraint has significant consequences on the quality and relevance of the data, especially in the case of people with severe deficiencies.

However, solutions can be considered to replace the measures with the use of regression equations [48] or musculoskeletal models [9,47,49,50]. One could thus consider determining maximum isometric torques and thus reduce the time required to evaluate several postures or an entire gesture. In addition, it could be interesting to verify the sensitivity of the models to variations in postures and joint couples of certain dofs.

In this context, the use of musculoskeletal models seems relevant to determine the FFS. Another important advantage of this type of formalism is the possibility of evaluating the FFSs regardless of the posture adopted. Indeed, the input data no longer come simply from joint torques, but directly from the isometric force generated by the muscles. Obviously, particular care will have to be taken to obtain an adequate model geometry and a scaling of the muscular forces in accordance with the physical capacities of an individual.

## 5. Conclusions

Four models (FE, SFE, FP and SFP) of force generation capacities were compared with force measurement (MFP). Several parameters were considered: isotropy, volume, maximum force, angle between principal axes and RMS errors. The proposed models gave rather good estimations of optimal force repartition and orientation, but amplitude information was evaluated less correctly. Further development should be considered by considering the multi-joint torque coupling due to musculoskeletal geometry. Despite some limitations, the proposed models could be of practical interest to estimate optimal force direction for applications in biomechanics, ergonomics and rehabilitation.

**Author Contributions:** Conceptualization, N.R., V.H. and P.G.; methodology, N.R., V.H. and P.G.; software, V.H.; validation, N.R., V.H. and P.G.; formal analysis, N.R., V.H. and P.G.; investigation, N.R., V.H. and P.G.; resources, P.G.; data curation, V.H.; writing—original draft preparation, N.R., V.H. and P.G.; writing—review and editing, N.R., V.H. and P.G.; visualization, V.H.; supervision, N.R. and P.G.; project administration, P.G.; funding acquisition, P.G. All authors have read and agreed to the published version of the manuscript.

**Funding:** This work was financially supported by a grant (6533-2013) from the Ministry of National Education (France).

**Institutional Review Board Statement:** The study was conducted according to the guidelines of the Declaration of Helsinki, and approved by the Ethics Committee of the University of Toulon.

**Informed Consent Statement:** Informed consent was obtained from all subjects involved in the study.

**Data Availability Statement:** The data presented in this study are available on request from the corresponding author..

**Conflicts of Interest:** The authors declare no conflict of interest.

## Abbreviations

The following abbreviations are used in this manuscript:

| | |
|---|---|
| FFS | Force feasible set |
| FE | Force ellipsoid |
| SFE | Scaled force ellipsoid |
| FP | Force polytope |
| SFP | Scaled force polytope |
| MFP | Measured force polytope |

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
