# Peer review of "Upper-Limb Isometric Force Feasible Set: Evaluation of Joint Torque-Based Models"

_2673-7078, doi:10.3390/biomechanics1010008_

Round 1
Reviewer 1 Report
The present study demonstrates four different methods to calculate the feasible force set as exerted from the end-point of a kinematic chain during a specific motion, based only on strict formulations of force and torque space given a single upper-limb posture. The authors conducted a thorough experimental procedure to measure force and joint torques of the end-effector in specific orientations/locations, to use them as ground truth data for comparisons with the calculations using the four methods explained in the methods section. The results showcase the limitations of the methods used, although they look promising as the methods can be used as surrogate to musculoskeletal modeling, especially for ergonomic purposes.
However, the language used for important terms in the paper are confusing and needs better refinement and explanation. Also, it is advisable to enrich the Abstract with more information on results and conclusions.
Specific comments:
Line 19: By isometric force, you refer to the force exerted at the end-effector? It can also be the maximum force of a muscle.
Lines 20-22: “Indeed, the capacity of a muscle to generate joint torque depends on its moment arm while the joint torque ability to produce force at the hand depends on the kinematic chain posture dependent Jacobian matrix. “
The meaning of this sentence is somewhat elusive. When you refer to joint torque, I suppose you refer to the torque caused by an external force or is the torque at the end-effector caused by the muscles in the kinematic chain? Please explain.
Lines 23 – 25:” Since the amplitude of the isometric forces depends on both posture and the direction of the force, the knowledge of a maximal value for a limited number of directions is not sufficient to represent globally the FFS.”
Which force do you mean in the statement “direction of the force” – muscle or external?
Do you mean that there is an infinite number of possible maximal isometric force values? Why a “limited number of directions is not sufficient”, since in your study you report that a FFS includes a non-infinite number of solutions. To my understanding, physiological constraints to joint motion impose a limited number of possible directions and consequently maximal muscle forces.
Line 29: When you refer to maximum joint torques, you mean the torques at the end of the kinematic chain?
Line 48: This is the first time you introduce the term dofs. Please give the full explanation first.
Line 59: FE is force ellipsoid but then you say force polytope. Please correct.
Line 210: Change the term “were involving” to were involved.
Lines 265-267: Not clear, especially the second part of the sentence. Please rephrase.
Line 268: Optimal in what sense?
Line 270: Naming joint torques as the source of musculoskeletal disorders is very vague. Please elaborate.
Line 275: Orientation of what? The principal axis?
Author Response
Reviewer 1
The present study demonstrates four different methods to calculate the feasible force set as exerted from the end-point of a kinematic chain during a specific motion, based only on strict formulations of force and torque space given a single upper-limb posture. The authors conducted a thorough experimental procedure to measure force and joint torques of the end-effector in specific orientations/locations, to use them as ground truth data for comparisons with the calculations using the four methods explained in the methods section. The results showcase the limitations of the methods used, although they look promising as the methods can be used as surrogate to musculoskeletal modeling, especially for ergonomic purposes.
However, the language used for important terms in the paper are confusing and needs better refinement and explanation. Also, it is advisable to enrich the Abstract with more information on results and conclusions.
Response: We wish to thank the reviewer for the insightful comments. The recommendations were thoroughly followed. The abstract has been modified to include information on results and conclusion.
The following sentences were added to the absctract
The Volume, shape and force prediction errors were assessed. The scaled ellipsoid underestimated the maximal mean force while the scaled polytope overestimated it. The scaled force ellipsoid underestimated the volume of the measured force distribution while that of the scaled polytope was not significantly different form the measured distribution but exhibited larger variability. All the models characterized well the elongated shape of the force distribution. The angles between the main axes of the modelled ellipsoids and polytopes and that of the measured polytope were compared. The values ranged from 7.3° for the scaled force ellipsoid to 14.3° for the force polytope. Over the entire surface of the scaled force ellipsoid, there were 39.7% of the points for which the prediction error was less than 50N, 33.6% for which the error was between 50 and 100N and 26.8% for which the error was greater than 100N with a maximum error of 244(79,1)N. For the force polytope the percentages were 56.2%, 28.3% and 15.4%, respectively and a maximum error of 305,6(85.5)N.
Specific comments:
- Line 19: By isometric force, you refer to the force exerted at the end-effector? It can also be the maximum force of a muscle.
Response: Indeed, you are right. We have replaced the term isometric by static In order to differentiate muscle and end effector force. The sentence was modified in the following way:
“Under static conditions, the distribution of the maximum external forces that a person can apply to the outside world, called force feasible set (FFS), is known to be anisotropic and posture-dependent. [10,11].”,
- Lines 20-22: “Indeed, the capacity of a muscle to generate joint torque depends on its moment arm while the joint torque ability to produce force at the hand depends on the kinematic chain posture dependent Jacobian matrix.“
The meaning of this sentence is somewhat elusive. When you refer to joint torque, I suppose you refer to the torque caused by an external force or is the torque at the end-effector caused by the muscles in the kinematic chain? Please explain.
Response: The sentence was not clear and was removed. We wanted to say that the dependence of the external force on the posture is due both to the muscular force and moment arm which contributes to the articular (joint) torques and to the Jacobian matrix of the upper limbs which links these articular torques to the external force at the end effector. The torques we are referring too are those created at the joints by the product of the muscular forces with their moment arm around the joints.
The sentence has been modified in the following way: “The dependence of the external force on the posture is due both to the muscular moment arm which contributes to the joint torques and to the Jacobian matrix of the upper limbs which links these joint torques to the external force at the end effector.”
- Lines 23 – 25: ”Since the amplitude of the isometric forces depends on both posture and the direction of the force, the knowledge of a maximal value for a limited number of directions is not sufficient to represent globally the FFS.” Which force do you mean in the statement “direction of the force” – muscle or external?
Response: We mean the direction of the external force. Indeed, the sentence lacks clarity because both the maximal muscular force and the external force depend on the posture. It has been replaced by the following one:
“Also, due to the anisotropy of the FFS, external forces amplitude can be very different according to the direction of these forces. Therefore, the knowledge of a maximal value for only one direction (or a limited set of directions) as it is usually found in the literature (cite ref) may be not sufficient to represent globally force capabilities of an individual at a given posture”
- Do you mean that there is an infinite number of possible maximal isometric force values? Why a “limited number of directions is not sufficient”, since in your study you report that a FFS includes a non-infinite number of solutions. To my understanding, physiological constraints to joint motion impose a limited number of possible directions and consequently maximal muscle forces.
Response: We wanted to say that since the maximum possible external force depend on external force direction and on the posture, a single value or a limited number of values of the maximal external force was not representative of the whole FFS. The modified sentence of comment 3 tries to emphasize this point.
- Line 29: When you refer to maximum joint torques, you mean the torques at the end of the kinematic chain?
Response: We meant the maximum torques applied at the joints due to muscle action, not at the upper-limb extremity.
- Line 48: This is the first time you introduce the term dofs. Please give the full explanation first.
Response: This was a mistake, the term dofs is detailed at its first appearance in the text.
- Line 59: FE is force ellipsoid but then you say force polytope. Please correct.
Response: Corrected
- Line 210: Change the term “were involving” to were involved.
Response: Corrected
- Lines 265-267: Not clear, especially the second part of the sentence. Please rephrase.
“The interest of such information is twofold [15,37,38]. In a motor task, if the direction of application of the measured forces is aligned with the main axis of the FFS model, it can be considered that this matter may adequately capture this aspect of the subject’s motor behaviour.”
Response: The response for this comment is combined with that of comment 10.
- Line 268: Optimal in what sense?
“In this eventuality, the application of a given level of force along the main FFS axis is optimal in some sense [15,38]”
Response: The authors agree with the comment.
The following text tries to answer to comments 9 and 10.
“The FFS provides information on the set of forces that the upper limb can exert on the environment. In particular, some directions allow for greater forces than others because of the anisotropy of the FFS. If for a given task corresponding to a given direction of force (e.g. drilling), the operator's posture causes the largest axis of the FFS to be aligned with the direction of the applied force, this means that the posture may be chosen to maximize the possible effort in the direction of interest. In other words, it would mean that the operator has chosen his posture in such a way that he can exert the greatest possible force given the capacities of his musculature. Moreover, the greater the force to be applied, the more the possible choices in terms of the combination of muscular forces should be reduced and the main axis of the FFS aligned with the force related to the task”
- Line 270: Naming joint torques as the source of musculoskeletal disorders is very vague. Please elaborate.
Response: The following sentence was added to clarify the link between joint troques and muculoskeletal disorders.
“The exertion of high intensity forces, handling heavy load over long period of time and working in unfavourable postures are main factors contributing to the incidence of musculoskeletal disorder. High joint torques may be associated with high muscular forces and increased joint reaction forces responsible for muscle and articular tissue damages leading to musculoskeletal disorders.”
- Line 275: Orientation of what? The principal axis?
“However, these two formalisms are those for which errors in orientation are the most important.”
Response: Yes indeed, the sentence has been modified in the following way
“However, these two formalisms are the ones for which the orientation errors of the principal axis with respect to the MFP main axis are the most important.”
Reviewer 2 Report
This manuscript compares four estimations of the hand isometric force feasible set for one given posture. Although this not clearly mentioned, these four estimations have been already extensively studied by the authors as well as evaluated with respect to the maximal isometric forces measured in multiple directions.
General comments
The introduction and discussion ignores several previous studies form the authors:
- “few studies that were only conducted in the horizontal plane” but Rezzoug et al. (2013) already extend this analysis to the 3D case;
- “or were based on one subject only” but Hernandez et al. (2016) studied 10 subjects with a very very similar material and method;
- “These results are original and constitute the first comparion of 3D modeling of FFS with both hand force and joint torque measurements on the upper limb” but Hernandez et al. (2016) already provided very similar results.
The method is based on the Jacobian matrix but surprisingly, no inverse kinematics is performed. “From the markers 3D coordinates, the joint angles were calculated in accordance with the ISB recommendations.” This means that the segment lengths (involved in the Jacobian) are not enforced to be constant. This also means that the 2 angles that should have been constrained at the elbow and wrist joint to have 7 degrees of freedom are neither enforced to be constant. This surely affects the other angles values and one may wonder if this can impact the final results or not.
“The joints coordinate definition and sequence of rotations around floating axes (shoulder ZXY, elbow ZY and wrist ZX) are consistent with the International Society of Biomechanics (ISB) recommendations.” This is not clear at all. In the joint coordinate system as recommended by the ISB, only one axis is said floating while the two others are embedded in the proximal and distal segments, respectively. When the joint angles are computed this way, it is always three angles about the three axes of the sequence of rotations that are computed. The mention to “elbow ZY and wrist ZX” sequence is very confusing and not consistent with the ISB.
Specific comments
The use of an OpenSim model for visualisation (Figure 8) brings confusion. The reader wonders if inverse kinematics was used or not (see general comments on this point). Moreover, it is not clear at all why the muscle wrapping objects are displayed.
“MCATAMMEY, L; NIGEL CORLETT, E. RULA: a survey method for the investigation of work-related upper limb disorders. Applied Ergonomics 1993.”
Why author’s names are in capitals?
No volume and page numbers provided: 24(2):91-9.
“Hernandez, V.; Rezzoug, N.; Jacquier-Bret, J.; Gorce, P. Human upper-limb force capacities evaluation with robotic models for ergonomic applications: effect of elbow flexion. Comput Methods Biomech Biomed Engin 2015, pp. 1–10. doi:10.1080/10255842.2015.1034117. ”
The date, volume, and pages are incorrect: 2016; 19(4):440-9.
Author Response
Reviewer 2
This manuscript compares four estimations of the hand isometric force feasible set for one given posture. Although this not clearly mentioned, these four estimations have been already extensively studied by the authors as well as evaluated with respect to the maximal isometric forces measured in multiple directions.
Response: We wish to thank the reviewer for the insightful comments. The recommendations were thoroughly followed.
General comments
- The introduction and discussion ignore several previous studies form the authors: “few studies that were only conducted in the horizontal plane” but Rezzoug et al. (2013) already extend this analysis to the 3D case;
Response: You are right, the study of 2013 presented preliminary results on one subject and not a complete set of parameters.
- “or were based on one subject only” but Hernandez et al. (2016) studied 10 subjects with a very very similar material and method.
Response: In this study the model’s predictions based on measured joint torques only were compared to each other’s for different elbow flexions. However, no ground truth measured forces obtained from a force sensor were used.
The following sentence was added in the introduction in response to comments 13 and 14.
For different elbow flexion values, a comparison between FFS predictions with FE, SFE, FP and SFP was done but no ground truth measures of maximal forces exerted at the hand was proposed ~\cite{Hernandez2016cmbbe}. Also, some preliminary results on one subject with both joint torques and 3D force measurements were provided in ~\cite{rezzoug2013}.
- “These results are original and constitute the first comparion of 3D modeling of FFS through FE, SFE, FP and SFP with both hand force and joint torque measurements on the upper limb” but Hernandez et al. (2016) already provided very similar results.
Response: Please see the response to the reviewer’s comment 14.
- The method is based on the Jacobian matrix but surprisingly, no inverse kinematics is performed. “From the markers 3D coordinates, the joint angles were calculated in accordance with the ISB recommendations.” This means that the segment lengths (involved in the Jacobian) are not enforced to be constant. This also means that the 2 angles that should have been constrained at the elbow and wrist joint to have 7 degrees of freedom are neither enforced to be constant. This surely affects the other angles values and one may wonder if this can impact the final results or not.
Response: The following paragraph was added to stress this point which was not detailed enough.
Upper-limb model section
“In order to compute the Jacobian matrix needed for FFS computation, the seven dofs kinematic chain model was defined through the Denavit-Hartenberg parameters with the sequence XZY for the shoulder, ZY for the elbow and XZ for the wrist. The elbow carrying angle (angle between the upper-arm and lower-arm in the sagittal plane in the anatomical reference posture) was integrated in the upper-limb model but not considered as a dof (Table \ref{tab:DH}). Given the model and the joint angles obtained from inverse kinematics (detailed in section \ref{section:expprotocol}), the kinematic chain Jacobian matrix was computed with the Matlab Robotics Toolbox~\cite{Corke2007}.”
Experimentla protocol
“{The length of the segments of the upper-limb model was obtained by computing the distance between the joint centers obtained from anatomical markers (middle of the radial and ulnar styloids for the wrist and middle of the medial and radial epicondyles for the elbow) and by the ScoRe method for the glenohumeral joint center~\cite{Ehrig2006,Monnet2007}. The elbow carrying angle, assessed from a static posture in the neutral position, was integrated in the upper-limb model but not considered as a dof. The position of the technical markers in the frame of the segment on which they were attached was computed from a static posture similar to that of the force measurements. Then using the upper-limb model and the technical markers, the joint angles were computed by the inverse kinematics algorithm proposed in~\cite{LU1999,ROUX20021279}”.
- “The joints coordinate definition and sequence of rotations around floating axes (shoulder ZXY, elbow ZY and wrist ZX) are consistent with the International Society of Biomechanics (ISB) recommendations.” This is not clear at all. In the joint coordinate system as recommended by the ISB, only one axis is said floating while the two others are embedded in the proximal and distal segments, respectively. When the joint angles are computed this way, it is always three angles about the three axes of the sequence of rotations that are computed. The mention to “elbow ZY and wrist ZX” sequence is very confusing and not consistent with the ISB.
Response : We thank you for this justified comment and this part has been clarified in the response to comment 16.
Specific comments
- The use of an OpenSim model for visualisation (Figure 8) brings confusion. The reader wonders if inverse kinematics was used or not (see general comments on this point). Moreover, it is not clear at all why the muscle wrapping objects are displayed.
Response: The authors agree with this comment, the Opensim representation has been removed for more clarity because it was not used and was there only for illustration. The inverse kinematics was performed but not with Opensim. A new figure has been used
- “MCATAMMEY, L; NIGEL CORLETT, E. RULA: a survey method for the investigation of work-related upper limb disorders. Applied Ergonomics 1993.” Why author’s names are in capitals?
Response: It was a mistake; this has been corrected
- No volume and page numbers provided: 24(2):91-9.
“Hernandez, V.; Rezzoug, N.; Jacquier-Bret, J.; Gorce, P. Human upper-limb force capacities evaluation with robotic models for ergonomic applications: effect of elbow flexion. Comput Methods Biomech Biomed Engin 2015, pp. 1–10. doi:10.1080/10255842.2015.1034117. ”
The date, volume, and pages are incorrect: 2016; 19(4):440-9.
Response: The reference has been updated.
Reviewer 3 Report
Interesting and well-designed study that incorporates both computational modeling and clinical testing. The manuscript is well-organized and provides a good amount of detail in most parts, but it is missing some main elements:
- The abstract and conclusion should include a summary of how well the models performed in comparison to one another. This is a very important outcome that needs to be presented more clearly.
- There needs to be a paragraph at the end of the Introduction section that introduces the measures that will be used to compare the models it MFP, along with an explanation of what they contribute to the performance of the model and a hypothesis of the outcome. This paragraph, or another, need to present an overview of the force and torque experiments and hypotheses regarding how they will relate to one another.
- Section 2 should explain how the experimental data was integrated with the computational model.
- All tables and figures need to be cited within the text prior to placement of the table or figure.
- The meaning of the angles in Table 2 needs to be more clearly described in lines 186-187. Explain the differences and their ramifications on the data.
- Acronyms need not be introduced in the abstract, but within the text, they should be defined a single time and then only the acronym should be used.
- FP needs to be defined in line 28 and then used in line 59.
- SFE and SFP are already defined in line 33, so they do not need to be redefined in line 60.
Further recommendations:
- The first paragraph set up the applications and importance of this study very well. However, I would like to see some statements regarding the upper limb in that paragraph, to further back-up the choice of the posture studied.
- A lot of space is used to define the angles used in the model, but they are not mentioned in the text with their angle names (theta 1-7). Since this is a Biomechanics journal, I recommend shortening the definition section of the angles so that just the biomechanical definitions are used – as this maps best with Table 2. For readers of this journal, Figure 1 seems unnecessary. I’d rather see the Jacobian matrix more clearly defined – where are the segment lengths integrated within it?.
- Define matrices R and T alongside equations (2) - (3) and (5) – (8), respectfully. Choose to present V with or without a subscript in equations (9) – (10), but not as both (unless there is a difference between the two, and then make that clear).
- Since all subjects are male, the use of “she” in line 123 is unnecessary and misleading.
- Acronyms need not be defined for the marker set since they are not used within the paper.
- The color scale in Figure 7 should be altered so that the gradient is clear even when printed in grey scale. As is, the scale goes from dark to light to dark in grey scale. Use only half the scale to make it easier for readers not able to see color.
- Double-check the formatting for references. The authors in first reference should not be in all caps.
Author Response
Reviewer 3
Interesting and well-designed study that incorporates both computational modeling and clinical testing. The manuscript is well-organized and provides a good amount of detail in most parts, but it is missing some main elements:
Response: We wish to thank the reviewer for the insightful comments. The recommendations were thoroughly followed.
- The abstract and conclusion should include a summary of how well the models performed in comparison to one another. This is a very important outcome that needs to be presented more clearly.
Response: The following sentences were added to the abstract.
The Volume, shape and force prediction errors were assessed. The scaled ellipsoid underestimated the maximal mean force while the scaled polytope overestimated it. The scaled force ellipsoid underestimated the volume of the measured force distribution while that of the scaled polytope was not significantly different form the measured distribution but exhibited larger variability. All the models characterized well the elongated shape of the measured force distributions. The angles between the main axes of the modelled ellipsoids and polytopes and that of the measured polytope were compared. The values ranged from 7.3° for the scaled force ellipsoid to 14.3° for the force polytope. Over the entire surface of the scaled force ellipsoid, there were 39.7% of the points for which the prediction error was less than 50N, 33.6% for which the error was between 50 and 100N and 26.8% for which the error was greater than 100N with a maximum error of 244(79,1)N. For the force polytope the percentages were 56.2%, 28.3% and 15.4%, respectively and a maximum error of 305,6(85.5)N.
- There needs to be a paragraph at the end of the Introduction section that introduces the measures that will be used to compare the models it MFP, along with an explanation of what they contribute to the performance of the model and a hypothesis of the outcome. This paragraph, or another, need to present an overview of the force and torque experiments and hypotheses regarding how they will relate to one another.
Response: Thank you for this suggestion that improves the introduction. The introduction has been modified in the following way.
“Different parameters will be considered in order to compare modelled FFSs and the MFP. They characterize the force amplitudes (maximum predicted force and volume which produces an overall evaluation) but also the shape of the FFSs (more or less elongated) and their orientation (angle between the main axes of the FFS). In addition, the RMS error is calculated for all points on the surface of the modelled FFS and the MFP. It is hypothesized that SFE will underestimate the MFP while the SFP will overestimate it thanks to their assumptions. It is also hypothesized that shape and orientation are correctly predicted by the models”
- Section 2 should explain how the experimental data was integrated with the computational model.
Response: How the experimental measure relates to the models construction and evaluation are detailed in section 2.5. Please see the response to the comment 16.
- All tables and figures need to be cited within the text prior to placement of the table or figure.
Response: This comment has been taken into account
- The meaning of the angles in Table 2 needs to be more clearly described in lines 186-187. Explain the differences and their ramifications on the data.
Response: A sentence has been added at the end of the “Data analysis” section and in the limitation at the end of the discussion.
Data analysis: The mean(SD) of joint angles displayed on the first line of Table \ref{tab:mesangles} correspond to that of the torques measurement on the dynamometer for the considered degree of freedom. The second line provides the mean(SD) of the joint angles of the complete upper-limb dofs during the force measurements.
Limitations : One of the limitations concerns the value of joint angles during the measurement of joint forces and torques. Indeed, the configuration of the dynamometer did not always allow to have joint angles of the unmeasured dofs very close to those adopted during the force measurement. For example, during shoulder torque measurements it was not possible to flex the elbow sufficiently. Since the maximum torques may depend on the position of adjacent joints this could have had an effect on the torque value. However, special care was given to having postures as close as possible between the two types of measures.
- Acronyms need not be introduced in the abstract, but within the text, they should be defined a single time and then only the acronym should be used.
Response: This has been corrected
- FP needs to be defined in line 28 and then used in line 59.
Response: Done
- SFE and SFP are already defined in line 33, so they do not need to be redefined in line 60.
Response: This point has been taken into account.
Further recommendations:
- The first paragraph set up the applications and importance of this study very well. However, I would like to see some statements regarding the upper limb in that paragraph, to further back-up the choice of the posture studied.
Response: We agree with the reviewer’s comment. The following text has been added to the experimental protocol section 2.5. “During the force measurements, the posture of the participant was standardized. The shoulder was slightly flexed an abducted with an elbow flexion of around 70°, a pronated forearm around 80°. The wrist was slightly extended. This posture is very common and corresponds to many situations in the workplace such as drilling, cart pushing for example. Also, this choice was dictated by the experimental set up with both joint torque and force measurements which was rather constraining and did not allow us to test many different postures..
- A lot of space is used to define the angles used in the model, but they are not mentioned in the text with their angle names (theta 1-7). Since this is a Biomechanics journal, I recommend shortening the definition section of the angles so that just the biomechanical definitions are used – as these maps best with Table 2. For readers of this journal, Figure 1 seems unnecessary. I’d rather see the Jacobian matrix more clearly defined – where are the segment lengths integrated within it?
Response : On order to respond to this comment, the Jacobian matrix calculation is more detailed in section 2.5 (response to comment 16).
- Define matrices R and T alongside equations (2) - (3) and (5) – (8), respectfully. Choose to present V with or without a subscript in equations (9) – (10), but not as both (unless there is a difference between the two, and then make that clear).
Response: The definitions have been added and Vell has been replaced by Volume for more clarity
The $[n\times n]$ matrix $\bm{V}$ is an orthogonal matrix whose columns form an orthonormal base of the torque space.
- Since all subjects are male, the use of “she” in line 123 is unnecessary and misleading.
Response: Corrected
- Acronyms need not be defined for the marker set since they are not used within the paper.
Response: The acronyms were removed
- The color scale in Figure 7 should be altered so that the gradient is clear even when printed in grey scale. As is, the scale goes from dark to light to dark in grey scale. Use only half the scale to make it easier for readers not able to see color.
Response: Thank you for this suggestion. The Figure 6 (former Figure 7) has been modified with a half scale.
- Double-check the formatting for references. The authors in first reference should not be in all caps.
Response: The references have been checked, corrected, and completed
Round 2
Reviewer 2 Report
The authors have clarified the reviewers' concerns about previous similar studies and about the methodology.
One of the modified sentence in the abstract can be further improved: "The dependence of the FFS on the posture is due to the muscle length which determine the muscle isometric force, to muscular moment arms which contribute to the joint torques, and to the Jacobian matrix of the upper limb which links these joint torques to the external force at the end effector."
Author Response
One of the modified sentence in the abstract can be further improved: "The dependence of the FFS on the posture is due to the muscle length which determines the muscle isometric force, to muscular moment arms which contribute to the joint torques, and to the Jacobian matrix of the upper limb which links these joint torques to the external force at the end effector."
Response : We would like to thank the reviewer for this suggestion, which has been included in the article (in blue).